# A Novel Mouse Model of Combined Hepatocellular-Cholangiocarcinoma Induced by Diethylnitrosamine and Loss of *Ppp2r5d*

**DOI:** 10.3390/cancers15164193

**Published:** 2023-08-21

**Authors:** Judit Domènech Omella, Emanuela E. Cortesi, Iris Verbinnen, Michiel Remmerie, Hanghang Wu, Francisco J. Cubero, Tania Roskams, Veerle Janssens

**Affiliations:** 1Laboratory of Protein Phosphorylation & Proteomics, Department of Cellular & Molecular Medicine, University of Leuven (KU Leuven), 3000 Leuven, Belgium; judit.domenechomella@kuleuven.be (J.D.O.); iris.verbinnen@kuleuven.be (I.V.); michiel.remmerie@kuleuven.be (M.R.); 2Translational Cell & Tissue Research, University of Leuven (KU Leuven), 3000 Leuven, Belgium; emanuelaelsa.cortesi@kuleuven.be (E.E.C.); tania.roskams@kuleuven.be (T.R.); 3Department of Immunology, Ophthalmology & ENT, Complutense University School of Medicine, 28040 Madrid, Spain; hangwu@ucm.es (H.W.); fcubero@ucm.es (F.J.C.); 4Health Research Institute Gregorio Marañón (IiSGM), 28007 Madrid, Spain; 5Centre for Biomedical Research, Network on Liver and Digestive Diseases (CIBEREHD), 28029 Madrid, Spain; 6Department of Pathology, University Hospitals Leuven (UZ Leuven), 3000 Leuven, Belgium; 7KU Leuven Cancer Institute (LKI), 3000 Leuven, Belgium

**Keywords:** hepatocellular carcinoma (HCC), combined hepatocellular-cholangiocarcinoma (cHCC-CCA), protein phosphatase 2A (PP2A), B56δ regulatory subunit, *Ppp2r5d*, diethylnitrosamine (DEN), mouse model, c-MYC, haploinsufficiency, tumor suppressor

## Abstract

**Simple Summary:**

Primary liver cancer (PLC) is among the leading causes of cancer-related deaths worldwide. PLC can be classified in hepatocellular carcinoma (HCC), cholangiocarcinoma (CCA), and the less common combined HCC-CCA (cHCC-CCA) based on histological features. The underlying mechanisms for PLC development/progression are still unknown, hampering the development of targeted PLC therapeutics. Protein Phosphatase 2A-B56δ (PP2A-B56δ) has been shown to have a tumor-suppressive role in mouse liver. We found that depletion of *Ppp2r5d* in mice accelerated HCC development induced by diethylnitrosamine (DEN), a well-established liver carcinogen, but unexpectedly also resulted in cHCC-CCA development. We also observed that *Ppp2r5d* is upregulated in tumors from wildtype and heterozygous mice, and that loss of *Ppp2r5d* alters specific oncogenes and signaling pathways in pre-tumor and tumor tissues. Our study highlights that mouse PP2A-B56δ has a tumor-suppressive role not only in HCC, but also in cHCC-CCA, which may have further implications for human PLC development and targeted treatment.

**Abstract:**

Primary liver cancer (PLC) can be classified in hepatocellular (HCC), cholangiocarcinoma (CCA), and combined hepatocellular-cholangiocarcinoma (cHCC-CCA). The molecular mechanisms involved in PLC development and phenotype decision are still not well understood. Complete deletion of *Ppp2r5d,* encoding the B56δ subunit of Protein Phosphatase 2A (PP2A)*,* results in spontaneous HCC development in mice via a c-MYC-dependent mechanism. In the present study, we aimed to examine the role of *Ppp2r5d* in an independent mouse model of diethylnitrosamine (DEN)-induced hepatocarcinogenesis. *Ppp2r5d* deletion (heterozygous and homozygous) accelerated HCC development, corroborating its tumor-suppressive function in liver and suggesting *Ppp2r5d* may be haploinsufficient. *Ppp2r5d*-deficient HCCs stained positively for c-MYC, consistent with increased AKT activation in pre-malignant and tumor tissues of *Ppp2r5d*-deficient mice. We also found increased YAP activation in *Ppp2r5d*-deficient tumors. Remarkably, in older mice, *Ppp2r5d* deletion resulted in cHCC-CCA development in this model, with the CCA component showing increased expression of progenitor markers (SOX9 and EpCAM). Finally, we observed an upregulation of *Ppp2r5d* in tumors from wildtype and heterozygous mice, revealing a tumor-specific control mechanism of *Ppp2r5d* expression, and suggestive of the involvement of *Ppp2r5d* in a negative feedback regulation restricting tumor growth. Our study highlights the tumor-suppressive role of mouse PP2A-B56δ in both HCC and cHCC-CCA, which may have important implications for human PLC development and targeted treatment.

## 1. Introduction

Primary liver cancers (PLC) in adults encompass two main histological subtypes, hepatocellular carcinoma (HCC) and the much less common intrahepatic cholangiocarcinoma (CCA). HCC and CCA are highly heterogeneous diseases in terms of their etiology, mutational landscapes, transcriptomes, and histological representation [1,2,3,4,5,6], and their initiation and progression mechanisms are still not completely understood [1,2,3]. All these aspects result in a lack of efficient targeted therapies and overall poor survival outcomes for liver cancer patients [4,5,6]. While HCCs generally arise from hepatocyte transformation [7,8,9,10], CCA is thought to derive from the intrahepatic cholangiocytes, hepatocytes, and/or liver bipotent progenitor cells [11,12,13]. Recent studies have identified a third subtype of PLC with tumors comprising both HCC and CCA morphological features, known as combined HCC-CCA (cHCC-CCA) or “biphenotypic” PLC [14,15,16]. Importantly, there are two main forms of cHCC-CCA: the classical form with tumors containing typical HCC and CCA areas, and the intermediate cell carcinoma with stem-cell features and composed of intermediate cells [17]. These heterogeneous tumors are more aggressive and have a poorer prognosis than HCC [16]. The cell of origin of cHCC-CCA tumors and the molecular mechanisms involved in phenotype decision are still highly controversial [18].

Protein Phosphatase 2A (PP2A) represents a family of cellular Ser/Thr-specific phosphatases, many of which have a proven tumor-suppressive function in different human tissues [19]. PP2A phosphatases are holoenzymes consisting of a catalytic C, a scaffolding A, and a regulatory B subunit, which broadly defines PP2A substrate specificity, function, and regulation of specific signaling pathways [20]. Inactivation of PP2A in cancer cells is a recurrent event that can be achieved by different mechanisms [21,22,23]. Most frequently, cancer-associated suppression of PP2A occurs by non-genomic alterations [24], such as increased expression of oncogenic cellular PP2A inhibitors, which are thought to inhibit the activity of specific tumor-suppressive PP2A holoenzymes [25]. Eventually, this inhibition results in increased growth and survival, via uncontrolled activation of several oncogenes, such as c-Jun, c-MYC, and β-Catenin, and/or oncogenic pathways, such as PI3K/AKT and MEK/ERK signaling [20,21,26], in particular, downstream of oncogenic or activated RAS [27,28,29,30].

In the liver, PP2A inactivation has been predominantly associated with HCC [31,32,33] and less frequently, with CCA [34]. However, the clinical picture remains largely incomplete and the impact of PP2A inhibition on the efficiency of (targeted) PLC therapies remains unresolved. In mouse models, the conditional knockout (KO) of PP2A Cα subunit in hepatocytes (*Alb*-Cre *Ppp2ca* KO) resulted in decreased steatosis and increased serum lipid levels, as well as in increased insulin signaling and glycogen storage [35]. These effects may in part be attributed to loss of the PP2A-B56γ complex [36]. When treated with carbon tetrachloride (CCl_4_), PP2A Cα-deficient livers were protected from chronic liver injury, correlating with impaired TGFβ/Smad2,3 signaling [37]. In *Ppp2r5d* KO mice, devoid of the regulatory PP2A-B56δ subunit in all tissues, increased spontaneous HCC development was observed [38]. Specifically, 17% of 12–18-month-old KO mice and 57% of 18–24-month-old KO mice developed HCC within a largely normal liver context. Conversely, age-matched wildtype (WT) mice did not display any spontaneous liver tumor formation [38]. Mechanistically, RNAseq analysis revealed a role for oncogenic c-MYC activation in the KO HCCs, further underscored by increased c-MYC Ser62 phosphorylation in all KO tumors, leading to increased c-MYC stability and expression [38]. Additional proteomics analysis of the B56δ KO HCCs provided independent evidence that *Ppp2r5d* KO mice represent a valuable model of hepatocarcinogenesis that captures many of the characteristics of the human disease [39].

Despite multiple efforts to develop mouse models to study HCC, none of the existing models have successfully captured all aspects of human PLC, including many of the typical genetic and cellular features or predisposing factors [40,41,42]. However, several studies focusing on expression profiles have shown that diethylnitrosamine (DEN)-induced hepatocarcinogenesis in mice resembles a subclass of human HCC associated with poor prognosis [43,44]. Consequently, this model is currently widely used for HCC studies in mice. The single administration of the DEN carcinogen into 14-day-old mice generates DNA adducts and leads to the formation of a reproducible mutagenic imprint in the hepatocytes, predominantly in *H-Ras* (50%), *B-Raf* (30%) and *Egfr* (20%) proto-oncogenes [42,45]. Thus, the DEN-induced mouse hepatocarcinogenesis model is particularly suitable for studying HCC development downstream of oncogenic RAS activation [46].

In the current study, we have applied the principles of DEN-induced hepatocarcinogenesis to *Ppp2r5d* KO mice to expedite tumor formation and further study the tumor-suppressive role of PP2A-B56δ in mouse liver. For this purpose, 2-week-old wildtype *Ppp2r5d +/+* (WT), heterozygous *Ppp2r5d* +/− (HE), and homozygous *Ppp2r5d* −/− (HO) mice were injected with DEN, and tumor development was followed up to 11 months post-DEN injection. Livers from mice at 6, 9, and 11 months post-DEN injection were macroscopically, histologically, and (immuno)histochemically characterized, and putative alterations in oncogenic signaling were analyzed by immunoblotting. Our results do not only show faster HCC development in both *Ppp2r5d* HO and HE strains, they also reveal the unusual development of cHCC-CCA tumors under these conditions. In addition, we show increased expression of *Ppp2r5d* in DEN-induced WT and HE tumors, but not in DEN-treated pre-malignant WT or HE livers, suggestive of the involvement of *Ppp2r5d* in a negative feedback regulation, downstream of a tumor-specific, oncogenic factor.

## 2. Materials and Methods

### 2.1. Experimental Mouse Models

The generation of constitutive, total body *Ppp2r5d* knockout mice (*Ppp2r5d* −/−) in a C57BL/6 background has been described before [47]. *Ppp2r5d* +/− (HE) mice were intercrossed to generate mice with two wildtype *Ppp2r5d* alleles (*Ppp2r5d* +/+ (WT)), or with knockout of one (*Ppp2r5d* +/− (HE)) or two *Ppp2r5d* alleles (*Ppp2r5d* −/− (HO)). Genotyping was performed by PCR on genomic DNA isolated from small ear pinches upon weaning (Forward: 5′-TACCACACGCTGTCTTCATC-3′, Reverse (WT): 5′-CACAGCACTGGCGTAGCTTC-3′, Reverse (KO): 5′-CGAAGCTTGGCTGGACGTAA-3′). Mice were bred and maintained in the Animal Facility at KU Leuven (Leuven, Belgium) under standard housing conditions and with ad libitum feeding and drinking. All animal procedures were approved by the KU Leuven Animal Ethical Committee (project P243-2015).

Male pups received 20 mg/kg (i.p.) of diethylnitrosamine (DEN, Sigma Aldrich, Hamburg, Germany, N0756) at 14 days of age and were sacrificed at 6, 9, and 11 months post-DEN administration by an overdose of Dolethal (100 mg/kg), followed by transcardial perfusion with phosphate-buffered saline (PBS, Sigma Aldrich, Gillingham, UK, D8537). Upon dissection, livers were removed, weighed, and macroscopically evaluated for lesions/nodules. The number of apparent nodules was documented, and each lesion was measured to calculate the ‘tumor area’ for each liver (= percentage of the total liver area that was affected by macroscopic lesions). Liver tissues were fixed in 4% paraformaldehyde (PFA, Sigma Aldrich, Hamburg, Germany, P6148) for histological analysis and immunohistochemistry (IHC) staining. For protein extraction purposes, healthy and tumor liver tissues were collected separately, snap-frozen in liquid nitrogen, and kept at −80 °C until further processing.

### 2.2. Histological Evaluation of Samples

Formalin-fixed and paraffin-embedded (FFPE) liver tissues were sectioned (5 μm) and stained for hematoxylin and eosin (H&E) and Sirius Red (SR) according to standard procedures, and evaluated in a blinded manner by an experienced liver pathologist (T.R.). Selected paraffin sections were subsequently used for additional IHC staining with different markers. Briefly, liver sections were deparaffinized with xylene and rehydrated with descendent percentage of ethanol. Sections were boiled for 10 min in Citrate Buffer pH 6 Antigen Retriever (Sigma-Aldrich, Newark, CA, USA, C9999), followed by 10 min of incubation with BLOXALL Blocking Solution (Vector Laboratories, Newark, CA, USA, SP-6000), incubation with 2.5% horse serum (Vector Laboratories, Newark, CA, USA, 30021) for 30 min, and overnight incubation at 4 °C with the primary antibody. The following primary antibodies were used: Ki67 (1/75, Abcam, Cambridge, UK, ab16667), CK19 (1/100, Abcam, Cambridge, UK, ab195872), EpCAM (1/100, Abcam, Cambridge, UK, ab71916), SOX9 (1/100, Abcam, Cambridge, UK, ab185966), and c-MYC (1/200, Abcam, Cambridge, UK, ab32072). The next day, slides were washed in PBS and incubated with anti-rabbit (Vector Laboratories, Newark, CA, USA, 30025) or anti-mouse (Vector Laboratories, Newark, CA, USA, 30027) HRP-conjugated secondary antibodies for 1 h at RT in a humidifying box. 3,3-diaminobenzidine solution (DAB, Vector Laboratories, Newark, CA, USA, 30215 and 30216) was used as a chromogen for antigen–antibody complexes. After IHC staining, slides were washed, counterstained with hematoxylin, and mounted using Roti-Mount (Carl Roth, Germany, HP68.1). Pictures were acquired with an Axio Imager A1 microscope (Carl Zeiss AG, Jena, Germany) equipped with AxioVision software (version 4.8.2, White Plains, NY, USA). Analysis was performed with ImageJ software (version 1.53t, Rasband, WS, USA)

### 2.3. Protein Extraction and Immunoblot Analysis

Frozen liver tissues were thawed on ice and homogenized (douncer) in 25 mmol/L Tris-HCl pH7.6, 150 mmol/L NaCl, 1 mmol/L EDTA, 1 mmol/L EGTA, supplemented with protease/phosphatase inhibitors (cOmplete™ protease inhibitor cocktail and PhosSTOP™ phosphatase inhibitor cocktail, Roche, Mannheim, Germany, 11836170001 and 04906837001) for 15 min on ice. After clearance (13,000× *g*; 20 min, 4 °C), supernatants were further analyzed.

For immunoblotting, proteins were separated by SDS-PAGE on 4–12% (*w*/*v*) Bis-Tris gels (Invitrogen, Carlsbad, CA, USA, WG1403BOX) and transferred to nitrocellulose membranes (Amersham^TM^ NC, Merck, Freiburg, Germany, 10600004). Membranes were blocked in 3% bovine serum albumin (BSA, Roche, Mannheim, Germany, 03116964001) in Tris-buffered saline (TBS)/0.1% Tween-20 for 1 h at room temperature (RT) and incubated with the primary antibody overnight at 4 °C. Primary antibodies (1/1000) were the following: B56δ (Abcam, Cambridge, UK, ab188323), B55α (Cell Signaling Technology, Danvers, MA, USA, 5689S), B56ε (in-house [48]), MST1 (Cell Signaling Technology, Danvers, MA, USA, 3682S), P(T183/T180)MST1/2 (Cell Signaling Technology, Danvers, MA, USA, 49332S), YAP (Cell Signaling Technology, Danvers, MA, USA, 14074S), P(S127)YAP (Cell Signaling Technology, Danvers, MA, USA, 13008S), AKT (Abcam, Cambridge, UK, ab179463), P(T308)AKT (Abcam, Cambridge, UK, ab38449), MEK1/2 (Cell Signaling Technology, Danvers, MA, USA, 9122S), P(S217/221)MEK1/2 (Cell signaling Technology, Danvers, MA, USA, 9154S), ERK1/2 (Cell Signaling Technology, Danvers, MA, USA, 9102S), P(T202/Y204)ERK1/2 (Cell Signaling Technology, Danvers, MA, USA, 9101S). After being washed briefly in TBS and 0.1% Tween-20, the membranes were incubated at RT for 1 h with secondary anti-mouse or anti-rabbit horseradish-peroxidase-conjugated antibodies (1/5000, Dako, Glostrup, Denmark, P02060 and Cell Signaling Technology, Danvers, MA, USA, 7074S) and developed on an ImageQuant LAS 4000 system (GE Healthcare, Uppsala, Sweden) using the WesternBright ECL detection kit (Advansta, San Jose, CA, USA, K-12045-D50). All densitometric quantifications were performed with Image Studio^TM^ Lite software (version 5, LI-COR, Bad Homburg, Germany, RRID: SCR_013715). Ponceau’s stains of the membranes were used for normalization.

### 2.4. Statistical Analysis

Data are expressed as mean ± standard error of the mean. Statistical significance was determined by performing a one-way or two-way ANOVA with Tukey’s multiple comparisons test, or Contingency of the Odd ratios with Fisher’s exact test using GraphPad Prism 8.0 Software (version 8, La Jolla, CA, USA)—as indicated. Values of *p* < 0.05 were considered significant.

## 3. Results

### 3.1. Analysis of Livers at 6 Months Post-DEN Injection Shows That Either Homozygous or Heterozygous Ppp2r5d Deletion Accelerates HCC Development

As explained, homozygous *Ppp2r5d* KO mice show a significantly increased predisposition to spontaneous HCC development upon aging [38]. To further assess the role of complete or partial *Ppp2r5d* loss on the initiation and/or progression of HCC in a chronic liver injury model, *Ppp2r5d* +/+ (WT), *Ppp2r5d* +/− (HE), and *Ppp2r5d* −/− (HO) male mice were injected with 20 mg/kg DEN at 2 weeks of age. Tumor formation was assessed at 6, 9, and 11 months post-DEN administration (Figure 1A). We did not include untreated mice in our aging cohort, as, in our previous study, we observed no signs of hepatic abnormalities or HCC development in male KO mice aged 6–12 months yet [38].

Although no significant differences in liver/body weight ratio between genotypes were observed at 6 months post-DEN treatment (Figure 1B), macroscopic and microscopic evaluation of the livers showed an overall increased apparent tumor burden in HE and HO mice compared to WT (Figure 1C–F). While only 25% of WT mice developed macroscopically visible lesions (nodules) in the liver, 43% of HE and 67% of HO mice showed signs of hepatocarcinogenesis (WT vs. HE: Odd ratio = 7 (*p* = 0.236); WT vs. HO: Odd ratio = 16.2 (*p* = 0.076); HE vs. HO: Odd ratio = 2.7 (*p* = 0.592)) (Figure 1C,D). These findings were further confirmed by quantifying the average number of macroscopic and microscopic nodules per liver in mice of a given phenotype (Figure 1E,F). HE and HO mice displayed a higher number of nodules per liver than WT mice (average number of nodules: WT = 0.25, HE = 1.86, HO = 3.5). Additionally, by measuring the average size (tumor area) of the lesions per liver, we found that HE and HO mice showed larger nodules than WT mice (average % tumor area: WT = 0.22, HE = 0.78, HO = 0.85) (Figure 1G).

To further evaluate the nature of the observed lesions at 6 months post-DEN injection, a detailed histopathological analysis was performed following H&E staining. As commonly seen in tumors derived from a single DEN injection [45], basophilic tumors with high nuclei/cytoplasmic ratios were present in all mice, regardless of their genotype. In WT mice, the single lesion observed in one mouse consisted of basophilic hepatocytes, with pale eosinophilic droplets, and paler ballooned cells with coarse inclusion bodies (Mallory–Denk-like bodies) (Figure 2A). All these features are characteristics of classical HCC. In HE mice, tumors with basophilic hepatocytes with a higher nuclei/cytoplasmic ratio were observed (Figure 2B). Moreover, eosinophilic pale round inclusions were seen in the cytoplasm of the basophilic cells (Figure 2B), contributing to the identification of additional small nodules. Tumor cells also showed some steatosis (Figure 2B). In HO mice, similar HCC tumors to those in HE mice were observed, but they were more numerous. Notably, one larger nodule showed a nodule-in-nodule appearance, indicating tumor progression (Figure 2C). This nodule-in-nodule was composed of an inner nodule with more clear hepatocytes displaying coarse eosinophilic inclusion bodies, surrounded by a larger outer basophilic cell nodule with eosinophilic pale inclusions. In another nodule in the HO mice, dilated sinusoids with recognizable hepatocytes in between these sinusoids were observed. Such a feature represents a condition known as ‘peliosis’ in human liver pathology.

In summary, both our macroscopic and histologic analyses of DEN-induced hepatocarcinogens in *Ppp2r5d* KO mice suggested that both HE and HO mice were more susceptible to HCC formation than WT mice. Such findings are underscored by the development of more, larger, and/or more progressed HCC tumors (e.g., nodule-in-nodule appearance) in HE and HO mice. Our data provide evidence that the loss of one *Ppp2r5d* allele is sufficient to accelerate HCC formation, thereby suggesting that *Ppp2r5d* may function as a haploinsufficient tumor suppressor gene.

### 3.2. Analysis of Livers at 9 and 11 Months Post-DEN Injection Highlights the Onset of a Combined HCC-CCA Phenotype Specifically in Homozygous and Heterozygous Ppp2r5d KO Mice

At 9 and 11 months post-DEN injection, macroscopic tumor formation was massive (i.e., 100%) in all three genotypes (Figure 3A). The tumor burden became exceedingly significant, to an extent that individual lesions could no longer be reliably counted or measured (Figure 3B—9 months, Figure 3C—11 months). Moreover, at 11 months, livers of the HE and HO mice showed a distinct macroscopic appearance, with features that remarkably resembled bile-filled structures and that were absent in the WT mice (Figure 3C).

To provide additional insights into potential differences in DEN-induced liver fibrosis or proliferation between genotypes at different stages post-DEN, Sirius Red and Ki67 stains were performed. Sirius Red stains showed overall increased fibrosis at 11 months as opposed to 9 and 6 months (6 m vs. 11 m WT *p* < 0.0001, HE *p* < 0.0001, HO *p* < 0.0001; 9 m vs. 11 m WT *p* = 0.003, HE *p* < 0.0001, HO *p* < 0.0001), a hallmark of DEN-induced liver damage [45,49]; however, no differences between genotypes were noted at a given age (Figure 3D). Thus, in the HE and HO mice, the increase in HCC development should not be considered as an indirect consequence of a different susceptibility of the mice to DEN-induced fibrosis. In contrast, Ki67 stains showed an increased number of proliferating cells in HE and HO mice as opposed to WT mice, particularly at 6 months (WT vs. HE *p* = 0.011; WT vs. HO *p* = 0.0194) (Figure 3E). This finding aligns with evidence of increased or more advanced tumor development in the HE and HO mice.

In WT livers, further histopathologic analysis disclosed similarities between tumors at 6 and 9 months post-DEN injection (classical HCC). However, tumors at 9 months were more numerous and larger in diameter. In some nodules, a nodule-in-nodule appearance was seen with the inner nodule composed of clear ballooned cells with coarse inclusion bodies. Additionally, some nodules exhibited very compact-growing hepatocytes or steatosis (Appendix A). At 9 months, analysis of the HE livers showed more and larger nodules compared to WT. Mostly, nodule-in-nodules with steatosis were seen (Appendix A). Intriguingly, beside HCCs, four of seven HE mice (43%) showed combined HCC-CCA (cHCC-CCA) tumors (WT vs. HE: Odd ratio = 4, *p* = 0.236), characterized by the presence of fibrous strands within the tumor showing ductules in continuity with small basophilic hepatocytes (Figure 4A,C). Occasionally, very small nodules already showed a cHCC-CCA phenotype. In cHCC-CCA tumors, ductules were more atypical and less structured if compared to the remaining portal tracts with ducts associated with limited ductular reaction (Figure 4A). The livers of 9 months post-DEN HO mice contained a higher number and larger nodules compared to HE mice. Similar to HE mice, the nodules showed a predominant nodule-in-nodule aspect. Additional nodule-in-nodules were composed of broad trabecular growing hepatocytes, as a sign of further progression of the tumors (Appendix A). In addition, 3/4 HO mice (75%) showed cHCC-CCA tumors with a morphology similar to that observed in the HE mice (WT vs. HO: Odd ratio = 6.7, *p* = 0.167) (Figure 4A,C).

To further characterize the cholangiocarcinoma component observed within the cHCC-CCA tumors from HE and HO mice at 9 months post-DEN injection, we performed additional immunohistochemical analysis. Many cells in the cholangiocarcinoma component stained positive for the biliary duct marker CK19, whereas the HCC component in the HE and HO tumors was negative for CK19 (Figure 5A). The transcription factor SOX9 has been reported to be expressed mainly in the hepatic progenitor cells (HPCs); however, in some cases, SOX9 can also be expressed in cholangiocytes and less differentiated hepatocytes [50]. Indeed, we observed that not only the cholangiocarcinoma compartment but also some of the HCC tumors in all three genotypes stained positive for SOX9, a marker associated with less differentiated hepatocytes (Figure 5A and Appendix A). Finally, staining for EpCAM, another HPC marker, further corroborated the cHCC-CCA phenotype seen in the HE and HO mice at 9 months post-DEN injection (Figure 5A). Remarkably, in some nodules from HE and HO mice, some hepatocytes also showed immunoreactivity for EpCAM, indicating an intermediate phenotype between hepatocytes and cholangiocytes (Appendix A).

The unusual development of cHCC-CCA tumors in DEN-treated HE and HO mice was further confirmed by histopathologic analysis of the liver tissues obtained at 11 months post-DEN injection. At 11 months, in WT livers, nodules were still recognizable as separate, non-confluent entities, mostly with nodule-in-nodule appearances. Occasionally, inner nodules showed a broad trabecular growth pattern with more nuclear atypia in tumor cells, or focal areas with a highly compact growth pattern. Overall, tumor progression was enhanced in mice at 11 months post-DEN injection compared to the 9 months post-DEN injection WT mice. However, no evidence was supportive for any cHCC-CCA development.

The HE livers at 11 months showed a higher number and larger tumors compared to those of the WT mice, although the tumors were still separate non-confluent entities, mostly with nodule-in-nodule appearances. Notably, all HE livers (10/10) displayed one or two cHCC-CCA tumors (WT vs. HE: Odd ratio = 99, *p* < 0.0001), with fibrous strands within the tumor where atypical ductules were present (Figure 4B,C). Bipotent progenitor cells were seen in continuity with these ductular structures, which exhibited positive staining for CK19 and EpCAM (Figure 5B). cHCC-CCA nodules showed immunoreactivity for SOX9 in the ductular cells and in many hepatocytes (Figure 5B and Appendix A). In the HO livers, a similar pattern to the one reported for HE livers was seen, but with larger and confluent nodules. Moreover, all HO mice (10/10) showed one or two cHCC-CCA tumors (WT vs. HO: Odd ratio = 99, *p* < 0.0001), with the ductular components staining positively for CK19, EpCAM, and SOX9 (Figure 4C and Figure 5B). Notably, at 11 months, 1/10 HO liver also contained a typical CCA (Figure 4D).

In summary, both our macroscopic and histopathological analysis of 9 and 11 months post-DEN HE and HO livers revealed striking differences compared to the WT controls, with apparent progressive development of cHCC-CCA in *Ppp2r5d*-deprived conditions (Figure 4). Of note, at 6 months post-DEN injection, some HE and HO mice additionally developed some ductular reaction at the edge of the portal tracts, indicative of bile duct proliferation (Appendix A). This phenomenon might be the result of increased ‘stress’ in the bile ducts, induced by the expansion of HCC nodule growth, which could precede the development of CCA-like features seen at the more advanced stages of the disease at 9 and 11 months post-DEN.

Overall, our analysis suggests a potential implication of PP2A-B56δ in suppressing the development of cHCC-CCA tumors upon DEN injection. Considering the suppression of B56δ expression in all liver cell types in HE and HO mice in this model and the ongoing controversy regarding the cell of origin of cHCC-CCA tumors [18,51,52], our data propose that *Ppp2r5d* may function either as a suppressor of biliary cell proliferation or as a (co-) determining factor in lineage commitment and cell fate of HPCs, in a chemically-induced carcinogenic context (see discussion).

### 3.3. DEN Treatment Induces Activation of MAPK, AKT, and YAP Pathways in Pre-Malignant Liver Tissues of WT, HE, and HO Mice, with AKT Activation Being Affected by (Partial) Loss of Ppp2r5d

To provide additional molecular insights into the role of (partial) *Ppp2r5d* depletion on DEN-induced signaling alterations in the liver, we examined the activation of three oncogenic signaling pathways induced downstream of oncogenic RAS. By using an immunoblotting approach (uncropped blots: Appendix A), we assessed the activation of the MAPK (ERK) pathway, the AKT pathway, and the YAP pathway, the latter as a result of an inactivation of Hippo signaling.

We first focused on the effects of DEN injection on the above-mentioned pathways in normal and pre-malignant liver tissues (i.e., before the onset of any tumor formation) in WT, HE, and HO mice. To this end, liver protein extracts were prepared from 2-month-old untreated WT, HE, and HO mice and from 6 months post-DEN treated mice of the same genotypes.

In agreement with previous reports [42], DEN administration induced the activation of MEK1/2 and ERK1/2 in WT livers, as evidenced by increased phosphorylation of MEK1/2 (Ser217/221) and ERK1/2 (Thr202/Tyr204) relative to total MEK1/2 and ERK1/2 levels (Figure 6A). In addition, there were no significant differences in MEK/ERK activation between WT, HE, or HO genotypes, although ERK activation appeared relatively diminished in the HO mice (Figure 6A).

Although AKT activity alterations have not frequently been reported in the DEN-induced mouse hepatocarcinogenesis model, abnormal activation of the PI3K/AKT signaling pathway is a hallmark of human HCC, and it has been associated with chronic liver injury [53]. Several reports have shown the direct regulation of AKT phosphorylation by PP2A [54,55,56], including the regulation through the PP2A-B56δ complex [57]. We found that DEN-injection induced a strong AKT activation in the livers of HE and HO mice, as evidenced by increased phosphorylation of AKT Thr308 relative to total AKT protein levels (Figure 6B). In the WT mice, AKT activation was less apparent, presumably due to technical reason, such as more background signal in the immunoblot in untreated conditions (Figure 6B). However, in DEN-treated conditions, we observed a trend towards higher AKT phosphorylation in the HE (*p* = 0.08) and HO livers (*p* = 0.09) as opposed to the WT livers, in concordance with a role of PP2A-B56δ in suppressing AKT activity/phosphorylation.

The tumor-suppressive function of the Hippo pathway in liver and how its dysregulation contributes to HCC pathogenesis have extensively been described [58]. Overexpression of the transcriptional co-activator Yes-associated protein (YAP), the major downstream target of the Hippo pathway, is an early event in DEN-induced hepatocarcinogenesis in rats and in human liver tumorigenesis [59]. In our mouse model, DEN treatment reduced phospho-MST1/2(Thr181/Thr180)/total MST1/2 levels in all three genotypes (Figure 6C), indicating an inactivation of the Hippo pathway upon DEN treatment. Accordingly, YAP protein expression was upregulated in all three genotypes, although most significantly in the WT mice (Figure 6C).

In summary, DEN-induced activation of MAPK and AKT signaling as well as inactivation of the Hippo pathway likely all contribute/predispose to eventual hepatocarcinogenesis in our models by promoting hepatocyte proliferation. Moreover, (partial) loss of B56δ did not seem to majorly affect these oncogenic signaling pathways, although a trend towards higher AKT activation was observed in the DEN-treated HE and HO livers compared to WT livers. This observation might corroborate a potential role for increased AKT activity in the earlier tumor onset seen in *Ppp2r5d* KO as compared to WT mice.

### 3.4. Only YAP, but Not MAPK or AKT Signaling, Is further Activated in Tumor versus Non-Tumor Tissue of DEN-Treated WT, HE, and HO Mice, with AKT and YAP Phosphorylation Being Affected by Ppp2r5d

To gain insights into the potential role of altered ERK, AKT, and Hippo signaling during tumorigenesis, we compared their activation in tumor versus non-tumor liver tissue of WT, HE, and HO mice at 11 months post-DEN injection. Additionally, we assessed how the (partial) loss of B56δ might further affect hepatocarcinogenesis through these pathways. Interestingly, we did not find major differences between MAPK or AKT activation between tumor and non-tumor tissues, regardless of the genotype of the mice (Figure 7A,B); just a modest activation of MEK1/2 and AKT was observed in HE and HO, and in WT respectively. In contrast, Hippo signaling was clearly further inactivated in the tumor versus non-tumor tissues in all three genotypes. Evidence of further increased YAP expression and decreased YAP Ser127 phosphorylation, a hallmark of increased nuclear and activated YAP [60], was observed (Figure 7C). These findings suggest a much larger contribution of DEN-induced YAP activation to liver tumorigenesis compared to DEN-induced ERK or AKT activation.

When comparing the three genotypes, no differences were seen in tumoral MAPK activation (Figure 7A); however, AKT activation was significantly higher in the HO mice, compared to WT and HE mice (Figure 7B). Notably, increased AKT activation was already observed in the non-tumor tissue and was further maintained in the tumors (Figure 7B), further substantiating a predisposing role for increased AKT activity in the earlier tumor onset seen in *Ppp2r5d* KO as compared to HE and WT mice. Moreover, YAP Ser127 phosphorylation levels were significantly lower in HE and HO mice, as compared to WT mice, both in the non-tumor and the tumor tissues (Figure 7C), thus corroborating an unexpected role for *Ppp2r5d* in suppressing YAP activation. These findings suggest that the loss of *Ppp2r5d* may also contribute to earlier tumorigenesis through the modulation of YAP activity.

Collectively, these results would hint to the involvement of the Hippo/YAP signaling pathway to tumor initiation/progression in the DEN-induced mouse hepatocarcinogenesis model. The complete or partial *Ppp2r5d* depletion further promoted YAP activation, whereas complete *Ppp2r5d* depletion resulted in the upregulation of AKT activity already in pre-malignant livers, which likely contributed to the increased tumor onset seen in HO versus HE *Ppp2r5d* KO mice.

### 3.5. Increased c-MYC Stabilization Contributes to the Liver Tumor Phenotype of HE and HO Mice

c-MYC is a well-known transcription factor involved in liver regeneration and hepatocarcinogenesis [61,62]. While for a long time, c-MYC overexpression has been associated with tumor progression and aggressiveness in HCC, recent studies have shown its implication in cHCC-CCA development [63]. The oncogenic potential of c-MYC is dependent on its gene (mRNA) expression as well as on distinct post-translational modifications, including phosphorylation and ubiquitination, that regulate c-MYC at protein level [64]. Several PP2A complexes have been described as essential regulators of c-MYC protein stability [65,66]. Specifically, PP2A-B56δ holoenzymes were found to decrease c-MYC protein stability by indirectly inducing c-MYC proteasomal degradation, via activation of GSK-3β, and its subsequent phosphorylation at Thr58 [67]. Moreover, in *Ppp2r5d* KO mice, spontaneous HCC development has been associated with increased oncogenic c-MYC expression and Ser62 phosphorylation [38]. Considering that activated AKT (Figure 6B and Figure 7B) may also contribute to increased c-MYC stability by suppressing GSK-3β activity [68,69], we hypothesized that c-MYC might contribute to the DEN-induced liver cancer phenotype observed in the *Ppp2r5d* HE and HO mice.

To further assess this possibility, we performed IHC analysis to determine c-MYC expression in all three genotypes at 6, 9, and 11 months post-DEN injection. While no c-MYC staining was seen in the HCCs from the WT mice at 6 months post-DEN (Figure 8A), the majority of the nodular HCC tumors were positive for c-MYC in HE and HO mice at this age (Figure 8B,C). These observations suggest a potential contribution of c-MYC expression to the earlier HCC onset in the HE and HO mice, as previously observed for spontaneous HCC development in the HO mice [38]. In contrast, at 9 and 11 months post-DEN, overall c-MYC staining did not show any obvious differences anymore between the three genotypes, as also WT HCC tumors stained positive for c-MYC.

### 3.6. Ppp2r5d Expression Is Upregulated in DEN-Induced Liver Tumors

In human cells, the *PPP2R5D* gene promotor is directly bound and transcriptionally activated by c-MYC in an E-box-dependent manner, providing a mechanism by which c-MYC limits its own abundance through a negative feedback involving increased PP2A-B56δ expression [67].

Interestingly, while verifying *Ppp2r5d* depletion in livers from WT, HE, and HO mice (Appendix A), we unexpectedly found altered expression of B56δ specifically in the tumor tissues as opposed to the non-tumoral surrounding tissues in both WT and HE mice (Figure 9A). Despite the existence of two mouse transcripts (NM_009358.3 and NM_001357684.1) encoding a 594 and 595-amino-acid protein of 72 kDa in the NIH Gene database and one transcript in the literature [38,70], we detected a second, faster migrating band in our WT and HE liver lysates, which likely corresponds to a proteolytic fragment of the full-length protein or an alternatively translated form, as it is no longer detected by the B56δ-specific antibody in the KO mice (Figure 9A). When combining and comparing the quantifications for both bands across samples, increased levels for total B56δ expression were observed specifically in the tumor in livers from WT and HE mice as opposed to the non-tumoral surrounding tissues (Figure 9A). This increase was specific for the PP2A B56δ subunit, as no changes were found for PP2A subunits B56ε or B55α (Figure 9B). In contrast, B56δ protein expression did not show any significant difference between pre-malignant livers at 11 months post-DEN treatment and untreated control livers for either WT or HE mice (Figure 9C), suggesting that the observed B56δ upregulation (Figure 9A) likely depends on a tumor-specific factor.

## 4. Discussion

In this study, we have further addressed the function of *Ppp2r5d*, encoding the PP2A regulatory B56δ subunit, in DEN-induced liver carcinogenesis. The DEN-induced hepatocarcinogenesis model is a well-established system for studying cancer development driven by mutant *H-Ras*, *B-Raf*, or *Egfr* in mice [42,71]. We demonstrated that complete as well as partial (50%) disruption of the *Ppp2r5d* gene in all liver cells resulted in earlier HCC onset. Surprisingly, we also observed the formation of cHCC-CCA tumors in older mice. Occasionally, CCAs were also observed in HO mice at 11 months post-DEN injection. Thus, while *Ppp2r5d*-depleted mice represented a model of spontaneous HCC arising in a normal liver context without obvious liver injury or inflammation [38], DEN-treated *Ppp2r5d* knockout mice represent a new model of cHCC-CCA, a type of PLC for which relatively few mouse models have been reported. Our data thus further confirm the tumor-suppressive role of PP2A-B56δ in the liver [38], and underscore for the first time that *Ppp2r5d* may actually exhibit haploinsufficiency for hepatocarcinogenesis.

Mechanistically, we showed that DEN injection resulted in the activation of the MAPK and AKT pathways downstream of activated RAS and in the inactivation of the Hippo pathway in pre-malignant livers. Although, in previous studies, the activation of the MAPK pathway has been associated with increased cell proliferation [72] and identified as a key oncogenic pathway in liver tumor development and progression [73,74,75,76,77], in our study, the MAPK pathway did not appear to be further activated in WT nor in tumors from HE and HO mice. These findings suggest that MAPK activation mainly predisposes for tumorigenesis, but that it is insufficient for actual tumor initiation in a DEN-induced mouse model. This observation concurs very well with studies in transgenic mouse models demonstrating that activating mutations in *H-Ras* alone are insufficient for inducing spontaneous liver tumor development, and require other oncogenic alterations, such as c-MYC overexpression [78,79].

The *c-MYC* oncogene has been reported as a driver gene for malignant conversion of pre-neoplastic liver lesions in human HCC [61], and for HCC tumor proliferation. In the current study, DEN-induced tumors from 6 months post-DEN injected HE and HO mice showed marked positivity for c-MYC, and presented with increased proliferation (Ki-67) compared to WT mice. Nevertheless, at later stages of the disease at 9 and 11 months post-DEN treatment, a subset of HCC tumors from WT mice were also positive for c-MYC, in line with the Ki67 staining. Previous studies have shown that c-MYC oncogenicity is post-translationally regulated, and that several PP2A complexes are implicated in this process [38,65,67,80]. Upon an oncogenic trigger, such as RAS/MEK/ERK activation, c-MYC becomes phosphorylated, increasing its oncogenicity [81]. However, PP2A-B56δ is capable of counteracting this event by activating glycogen synthase kinase-3β (GSK-3β), which eventually induces c-MYC proteasomal degradation [67]. In agreement with this mechanism, our data confirm the role of c-MYC in HCC initiation, and suggest that c-MYC is likely contributing to the earlier onset of DEN-induced HCCs observed in HE and HO mice at 6 months post DEN-injection. In contrast, AKT, another common hallmark of human HCC, triggers the inhibitory phosphorylation of GSK-3β, leading to increased stability and accumulation of c-MYC and other oncogenes, such as c-JUN and β-catenin [78,79]. As previously shown, AKT activity is also dependent on its phosphorylation status, which is known to be regulated by PP2A-B56δ in some models [57]. Although AKT overexpression alone can result in HCC formation with a long latency in mice [82], the co-expression of AKT and N-Ras has been described to accelerate HCC and CCA development through mechanisms that involve c-MYC activation [83]. Notably, we observed that the activation of AKT in pre-malignant liver tissues from HE and HO mice at 6 months post-DEN, appeared to be further increased in HO versus HE tumors at 11 months post-DEN. Our findings may imply that *Ppp2r5d* depletion could result in increased c-MYC activation through AKT activation, thus contributing to increased tumorigenesis and tumor progression. Moreover, our data suggest that deletion of both *Ppp2r5d* alleles would be necessary to achieve maximal AKT activation.

In human cells, c-MYC induces *PPP2R5D* expression in an E-box-dependent manner [67], thus constituting a negative feedback loop between PP2A-B56δ and c-MYC that restricts c-MYC oncogenicity. Cunningham et al. have reported *PPP2R5D* amplifications in several human cancers, including colon adenocarcinoma and HCC [84]. In our study, we found upregulation of B56δ protein expression in WT and HE mice, specifically in the DEN-induced tumors but not in the pre-malignant livers, suggesting that a tumor-specific factor is involved in the process. Based on analogous findings in human data [67], we speculate that this factor may likely be c-MYC. In support of this hypothesis, our in silico analysis of the mouse *Ppp2r5d* gene promotor identified two canonical E-boxes (CACGTG) at nt -388 and nt -1604 (relative to the presumed transcription initiation site) and ten non-canonical E-boxes (CANNTG) between nt -1 and nt -1650 (Appendix A). Whether any of these E-boxes are indeed involved in the transcriptional regulation of *Ppp2r5d* by c-MYC and in the upregulation of B56δ expression in the DEN-induced tumors remains to be determined.

Likewise, in vivo c-MYC driven hepatocarcinogenesis also depends on additional genetic alterations and/or oncoproteins [85]. For instance, in c-MYC-induced HCC tumors, TAZ depletion results in tumor regression [86], corroborating a facilitating role of YAP activation in c-MYC-induced hepatocarcinogenesis. In pre-malignant hepatocytes, the YAP overexpression is known to have a tumor-suppressive role, whereas its overexpression has a tumor promotor effect in HCC tumors [87]. Importantly, YAP and TAZ expression are essential to maintain liver tumors [87]. YAP stability is post-translationally regulated by the upstream kinases LATS1/2 and MST1/2 from the Hippo pathway. Phosphorylation of YAP at Ser127 results in its retention in the cytosol, while YAP phosphorylation at Ser909 promotes its proteasomal degradation. In line with these findings, pre-malignant livers from 6 months DEN-treated mice showed YAP overexpression in WT mice, and, to a lesser extent, in HE and HO mice. Furthermore, YAP protein expression was upregulated in 11 months post-DEN tumor lysates. In the pre-malignant livers from 11 months post-DEN HE and HO mice, YAP displayed enhanced activity as shown by decreased Ser127 phosphorylation, and became further activated in tumors from all genotypes. These data suggest for the first time a role for PP2A-B56δ in the regulation of YAP activity, providing an additional explanation for accelerated tumor progression upon loss of *Ppp2r5d*.

*Ppp2r5d* depletion did not only result in accelerated tumorigenesis, but also in the development of cHCC-CCA tumors at a later stage of the disease. Interestingly, several studies have shown the relevance of c-MYC in developing the cHCC-CCA phenotype. Similar to our mouse model, difficulties in establishing the relation between c-MYC and cHCC-CCA were reported in other studies. For instance, DEN treatment in hepatocyte-specific HNF4α KO mice resulted in the development of cHCC-CCA [88], and showed induction of c-MYC expression in both tumors and surrounding tissues. In another publication, the depletion of a subunit of the IKK complex, called NEMO, resulted in accelerated MYC-driven carcinogenesis in hepatocytes and in a c-MYC–dependent phenotypic transition from HCC to cHCC-CCA [63]. In their work, Liu and colleagues have shown that the co-expression of H-Ras/Myc in hepatocytes of *p53*+/− and *p53*−/− mice gave rise to cHCC-CCA tumors [79]. In a mouse model of cholangiocarcinogenesis, tumorigenesis was impaired after c-MYC deletion [89]. Nonetheless, opposite results from a recent study were showing a role for c-MYC in lineage commitment in *K-Ras*-driven primary liver cancer development [90]. Altogether, these studies suggest that c-MYC may play different roles in cHCC-CCA development, depending on the oncogenic trigger and the type of liver injury [51].

In human patients and mouse models [91], tumors with mixed HCC and CCA features (combined PLCs) are more aggressive than typical HCCs, characterized by tumor cells with high plasticity and stemness properties. Despite findings on their stem-like phenotype, the cell of origin of cHCC-CCA still remains highly controversial [51]. Indeed, although several studies have been hinting towards the contribution of hepatic progenitor cells (HPCs) in the development of cHCC-CCA tumors, other studies have suggested that de- or transdifferentiation of HCC-like hepatocytes may be the source of the CCA component in cHCC-CCA tumors. In line with studies from others [63], in our mouse model, the CCA tumor component within the cHCC-CCA tumors from the HE and HO mice stained positive for the progenitor markers SOX9 and EpCAM.

Interestingly, PI3K/AKT pathway can play a role in determining the fate of tumor cells under *K-Ras* oncogenic activation [92]. The depletion of *Pten*, a negative regulator of PI3K/AKT pathway, together with *K-Ras* activation in progenitor cells results in CCA tumors. In addition, the depletion of *Pten* in SOX9+ cells gives rise to cHCC-CCA tumors only in the background of hepatic injury in adult livers [93]. Thus, activation of AKT upon *Ppp2r5d* depletion, together with DEN treatment, could explain the induction of the cHCC-CCA tumors. YAP has also been reported to be implicated in the development of cHCC-CCA tumors, with an expansion of SOX9+ BEC-like cells [50]. Specifically, co-expression of the YAP activating mutant (S127A) and a constitutive mutant of PI3K in hepatocytes induces HCC, CCA and cHCC-CCA [94].

SOX9 is mainly expressed in biliary epithelial cells (BEC) [50] in the adult liver; however, in the presence of liver injury, SOX9 expression appears in hepatic progenitor cells and in hepatocytes around damaged bile ducts, resulting in less differentiated hepatocytes with biliary features [95]. We found SOX9-positive hepatocytes in tumors from 9 and 11 months post-DEN mice regardless of the genotype, indicative of less differentiated hepatocytes. In contrast, these hepatocytes were negative for EpCAM and CK19, cholangiocytes, or stem/progenitor cells markers. Furthermore, morphological characterization of these hepatocytes suggests these tumors are undifferentiated HCCs [17]. Indeed, upon chronic injury, a population of atypical ductular cells behaving as bipotent progenitor cells, originating from the bile ducts, are involved in liver repair [96,97]. Due to their characteristics, bipotent progenitor cells proliferate and differentiate into hepatocytes and biliary cells. Along these lines, the hyperproliferation of SOX9+ cells, i.e., progenitor cells, that we observed in HE and HO mice could also reflect the degree of liver damage that these mice undergo, after a combination of DEN treatment and *Ppp2r5d* depletion.

YAP and SOX9 expression are associated with tumor progression and poor prognosis and survival in patients with HCC and CCA [98,99], suggesting a potential role as predictive and prognostic markers in patients with PLC. As for many other transcription factors, such as YAP and c-MYC, SOX9 expression—and therefore activity—can be regulated at a post-translational level. While upon AKT-mediated phosphorylation, SOX9 activity increases [100], phosphorylation mediated by GSK-3β results in SOX9 proteasomal degradation [101,102], similarly to c-MYC. Such evidence could therefore imply that PP2A-B56δ holoenzyme may be involved in the regulation of SOX9 proteasomal degradation, via AKT inhibition and GSK-3β activation, thereby contributing to the development of cHCC-CCA tumors. Unfortunately, our mouse model does not allow determining the cell of origin of the cHCC-CCA tumors in HE and HO mice, and thereby potentially uncover a role of *Ppp2r5d* in liver cell plasticity upon injury. Additional experiments will be required to address this question in hepatocyte-, cholangiocyte-, or progenitor cell-specific *Ppp2r5d* KO mice.

## 5. Conclusions

We independently confirmed the tumor-suppressive role of the PP2A-B56δ holoenzyme in liver using the well-established DEN-induced mouse model of hepatocarcinogenesis. Remarkably, we found that both complete as well as monoallelic (partial) loss of *Ppp2r5d* not only accelerated DEN-induced HCC development, but also resulted in the unusual development of combined HCC-CCA tumors in this model, at a later stage of the disease. Mechanistically, we discovered that—partial—loss of *Ppp2r5d* promoted AKT and YAP activation in pre-malignant liver as well as liver tumor tissue in DEN-treated mice. Moreover, c-MYC expression was found specifically co-upregulated in the tumors, likely resulting in more proliferative and aggressive lesions. We hypothesize that PP2A-B56δ might be involved in the regulation of SOX9 protein stability and that, in hepatocarcinogenesis, PP2A-B56δ dysregulation could result in the upregulation of the oncogenes involved in progenitor cell induction/hepatocyte dedifferentiation and cell fate. Further studies will be needed to corroborate the role of PP2A-B56δ in liver tumor phenotype decision.

## Figures and Tables

**Figure 1 cancers-15-04193-f001:**
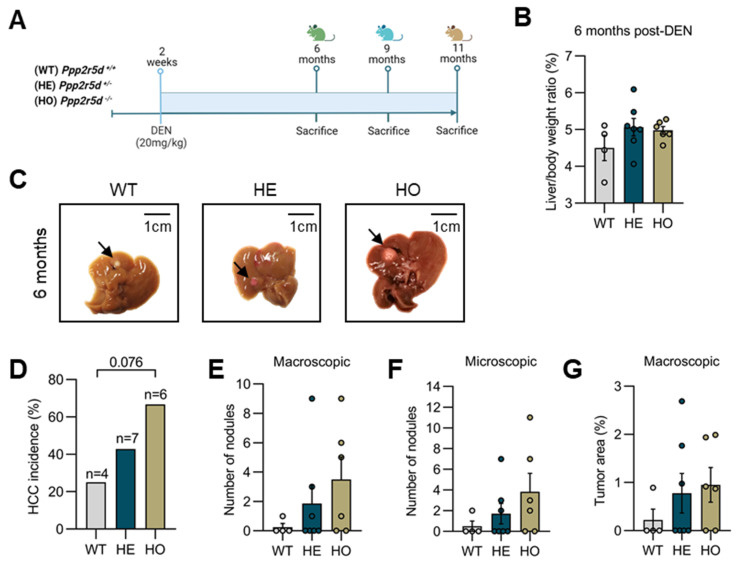
Macroscopic and microscopic liver analysis of WT, HE, and HO *Ppp2r5d* KO mice at 6 months post-DEN treatment. (**A**) Schematic diagram of experimental design. Two-week-old *Ppp2r5d* +/+ (WT), *Ppp2r5d* +/− (HE), and *Ppp2r5d* −/− (HO) male mice were injected with DEN (20 mg/kg) and sacrificed after 6, 9, and 11 months. (**B**) Mean liver weight relative to total body weight (%). (**C**) Representative images of livers from WT (*n* = 4), HE (*n* = 7), and HO (*n* = 6) mice treated with DEN (20 mg/kg), highlighting the macroscopic lesions (arrows). (**D**) HCC incidence in WT vs. HE; WT vs. HO; and HE vs. HO mice. The calculated odd ratio to develop HCC in WT vs. HE is 7 (*p* = 0.236); in WT vs. HO is 16.2 (*p* = 0.076); in HE vs. HO is 2.7 (*p* = 0.592). (**E**) Quantification of the number of macroscopic nodules per liver in the three genotypes. (**F**) Quantification of the number of microscopic nodules per liver in the three genotypes. (**G**) Macroscopic assessment of the tumor area (%) per liver in each of the three genotypes. Dots represent individual measurements within a group. Data are represented as mean ± S.E.M. One-way ANOVA and Contingency of the Odd ratios analysis were used. GraphPad Prism 8.0.

**Figure 2 cancers-15-04193-f002:**
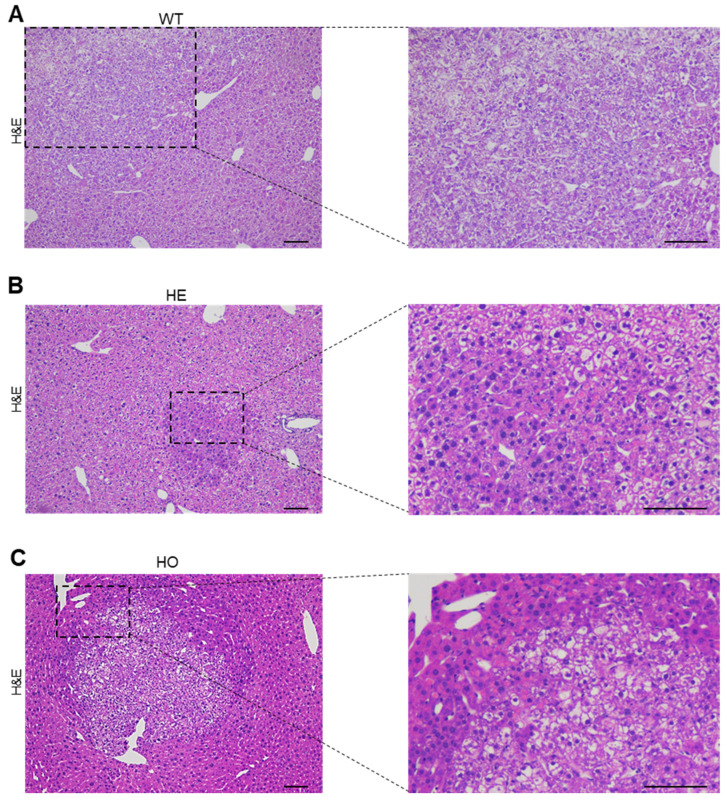
Histologic liver analysis of WT, HE, and HO *Ppp2r5d* KO mice at 6 months post-DEN treatment. (**A**–**C**) Representative images of liver sections of 6 months post-DEN WT (**A**), HE (**B**), and HO (**C**) mice stained for H&E. The right panel corresponds to the amplified area within the dotted lines of the left panel. Scale bar represents 100 μm.

**Figure 3 cancers-15-04193-f003:**
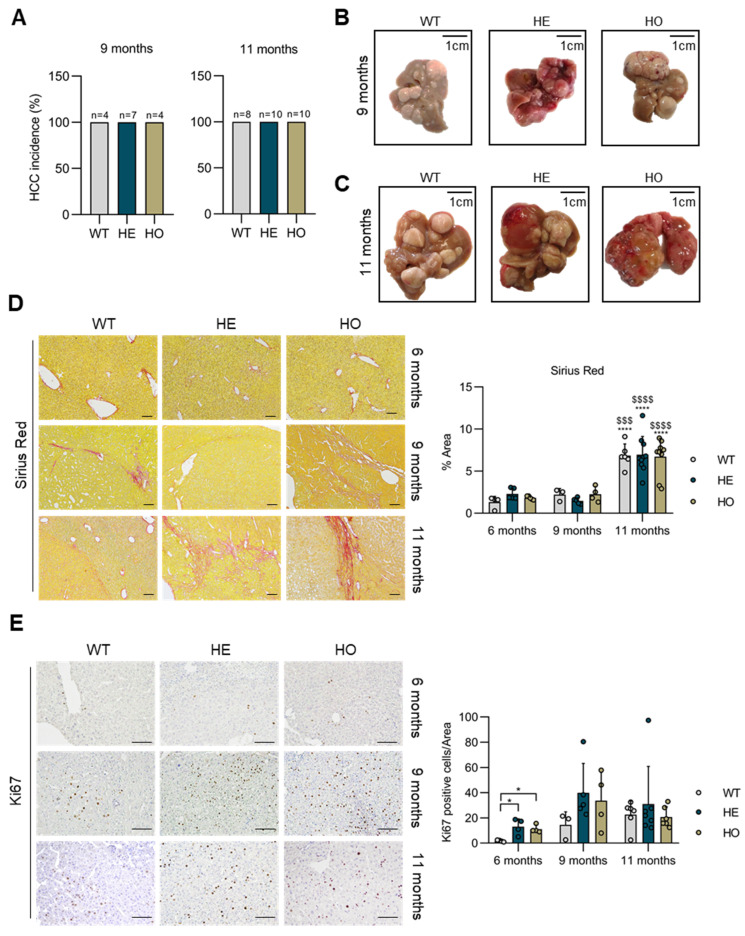
Macroscopic liver analysis of WT, HE, and HO *Ppp2r5d* KO mice at 9 and 11 months post-DEN treatment, and immunohistochemical analysis for Ki67, and Sirius Red staining in livers from 6, 9, and 11 months post-DEN treatment. (**A**) All WT (*n* = 4; *n* = 8), HE (*n* = 7; *n* = 10), and HO (*n* = 4; *n* = 10) mice showed macroscopic tumor lesions at 9 and 11 months post-DEN treatment. (**B**,**C**) Representative images of livers from 9 months (**B**) and 11 months (**C**) post-DEN WT, HE, and HO mice; (**D**) Microscopic analysis of DEN-induced liver fibrosis in the function of time and genotype visualized by Sirius Red staining. Representative images are shown (left). Quantification of the Sirius Red positive area (%) was performed using ImageJ (right). Scale bar represents 100 μm. (**E**) Immunohistochemical analysis of cell proliferation in function of time and genotype visualized by Ki-67 staining. Representative images are shown (left). Quantification of Ki67 positive cells (right). Scale bar represents 100 μm. Dots represent individual measurements within a group. Data are represented as mean ± S.E.M. (* *p* < 0.05, **** *p* < 0.0001, ^$$$^ *p* < 0.001, ^$$$$^ *p* < 0.0001). One-way and two-way ANOVA analysis were used. GraphPad Prism 8.0.

**Figure 4 cancers-15-04193-f004:**
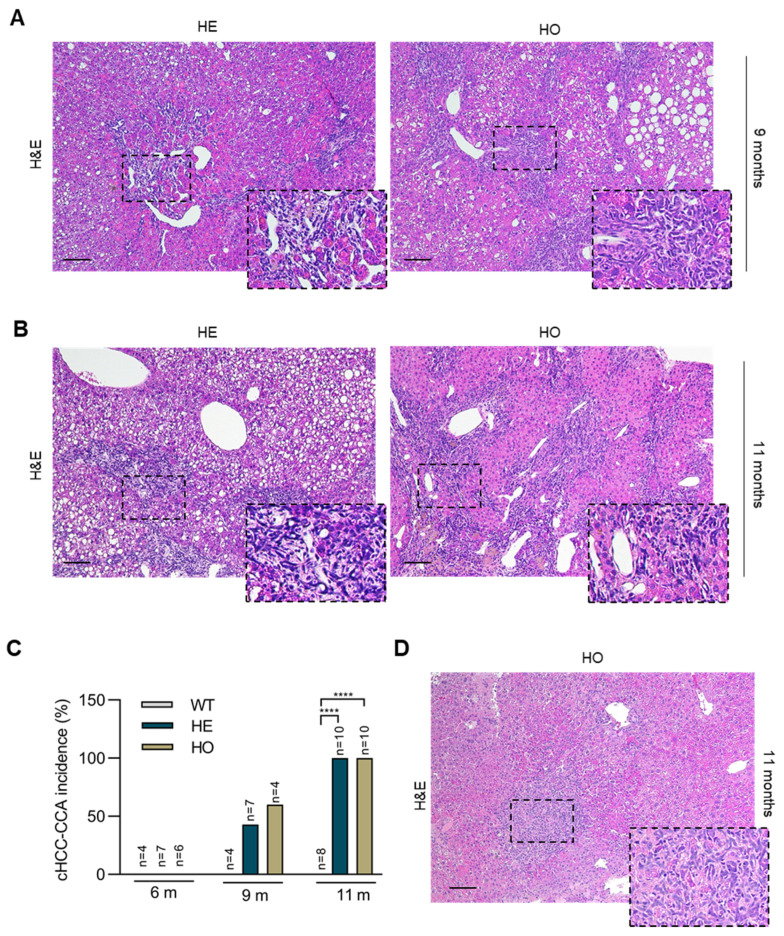
Histologic liver analysis of WT, HE, and HO *Ppp2r5d* KO mice at 9 and 11 months post-DEN treatment shows the development of cHCC-CCA tumors in HO and HE mice. (**A**,**B**) Representative images of liver sections of 9 months (**A**), and 11 months (**B**) post-DEN HE and HO mice stained for H&E. Area within the dotted lines shows the CCA component in the HCC tumors. (**C**) cHCC-CCA incidence in WT vs. HE, WT vs. HO, and HE vs. HO mice (%). The calculated odd ratio to develop cHCC-CCA in WT vs. HE is 4 (*p* = 0.236); in WT vs. HO is 6.7 (*p* = 0.167); in HE vs. HO is 1.3 (*p* > 0.999) at 9 months post-DEN; in WT vs. HE is 99 (*p* < 0.0001); in WT vs. HO is 99 (*p* < 0.0001) at 11 months post-DEN. (**D**) Representative image of a CCA tumor in a liver section of an 11-month-old post-DEN HO mouse stained for H&E. Scale bar represents 100 μm. (**** *p* < 0.0001). Contingency of the Odd ratios analysis was used. GraphPad Prism 8.0.

**Figure 5 cancers-15-04193-f005:**
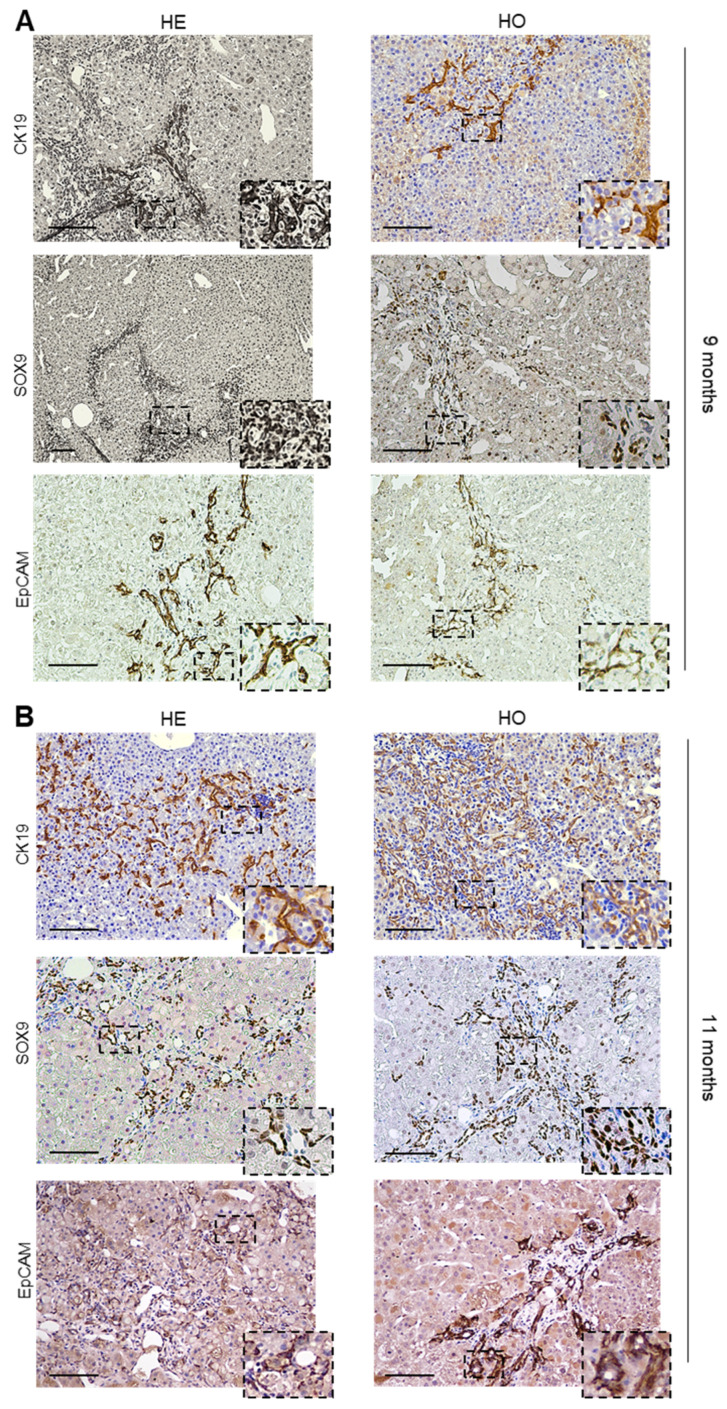
Characterization of the cHCC-CCA tumors from WT, HE, and HO *Ppp2r5d* KO mice at 9 and 11 months post-DEN. (**A**,**B**) Representative images of liver sections of 9 months (**A**), and 11 months (**B**) post-DEN HE and HO mice immunohistochemically stained for CK19, SOX9, and EpCAM. The area within the dotted lines shows the CCA component positive for CK19, SOX9, and EpCAM markers in the HCC tumors. Scale bar represents 100 μm.

**Figure 6 cancers-15-04193-f006:**
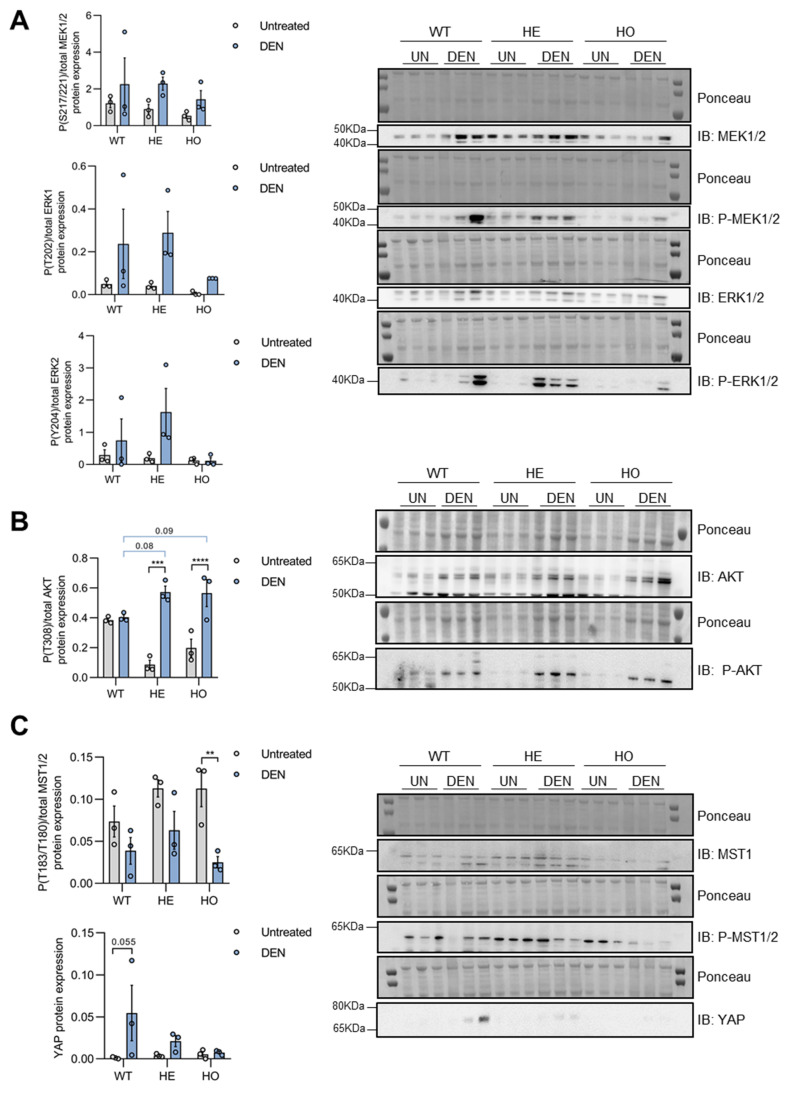
Analysis of MAPK, AKT, and Hippo pathway activation by DEN-treatment in WT, HE, and HO *Ppp2r5d* KO mice at 6 months post-DEN treatment. Protein extracts were prepared from untreated livers (UN) of WT, HE, and HO mice (2 months of age), as well as from the non-malignant (=not yet cancerous) tissue of DEN-treated livers (DEN) of WT, HE, and HO mice (at 6 months post-DEN injection). Equal amounts of lysate were separated by SDS-PAGE and subjected to immuno-blotting with the indicated antibodies. Ponceau was used for normalization. (**A**) MAPK pathway activation was assessed by determining MEK1/2 (Ser217/221) and ERK1/2 (Thr202/Tyr204) phosphorylation. (**B**) AKT pathway activation was assessed by determining AKT (Thr308) phosphorylation. (**C**) Hippo pathway activation was assessed by determining MST1/2 (Thr183/Thr180) phosphorylation and total YAP expression. Dots represent individual measurements within a group. Data are represented as mean ± S.E.M (** *p* < 0.01, *** *p* < 0.001, **** *p* < 0.0001). Two-way ANOVA analysis was used. GraphPad Prism 8.0. See Appendix A for the original image of Western Blots.

**Figure 7 cancers-15-04193-f007:**
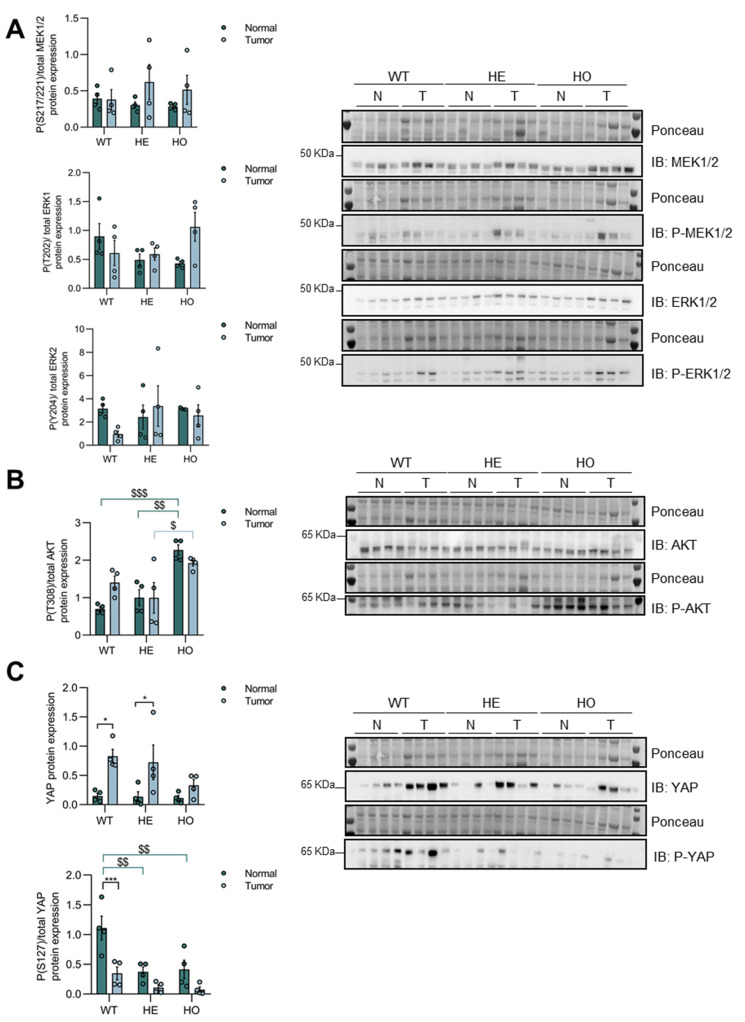
Analysis of MAPK, AKT, and Hippo pathway activation in tumor versus non-tumor tissues of livers from WT, HE, and HO *Ppp2r5d* KO mice at 11 months post-DEN treatment. Protein extracts were prepared from DEN-treated livers of WT, HE, and HO mice (11 months post-DEN), encompassing tissue parts that macroscopically still appeared healthy (= non-tumoral, N), or encompassing tissue that was clearly cancerous (= tumors, T). Equal amounts of lysate were separated by SDS-PAGE and subjected to immunoblotting with the indicated antibodies. Ponceau was used for normalization. (**A**) MAPK pathway activation was assessed by determining MEK1/2 (Ser2177/221) and ERK1/2 (Thr202/Tyr204) phosphorylation. (**B**) AKT pathway activation was assessed by determining AKT (Thr308) phosphorylation. (**C**) Hippo pathway activation was assessed by determining total YAP expression and YAP (Ser127) phosphorylation. Dots represent individual measurements within a group. Data are represented as mean ± S.E.M. (* = normal vs. tumor within a genotype; ^$^ = comparison between genotypes) (* *p* < 0.05, *** *p* < 0.001, ^$^ *p* < 0.05, ^$$^ *p* < 0.01, ^$$$^ *p* < 0.001). Two-way ANOVA analysis was used. GraphPad Prism 8.0. See Appendix A for the original image of Western Blots.

**Figure 8 cancers-15-04193-f008:**
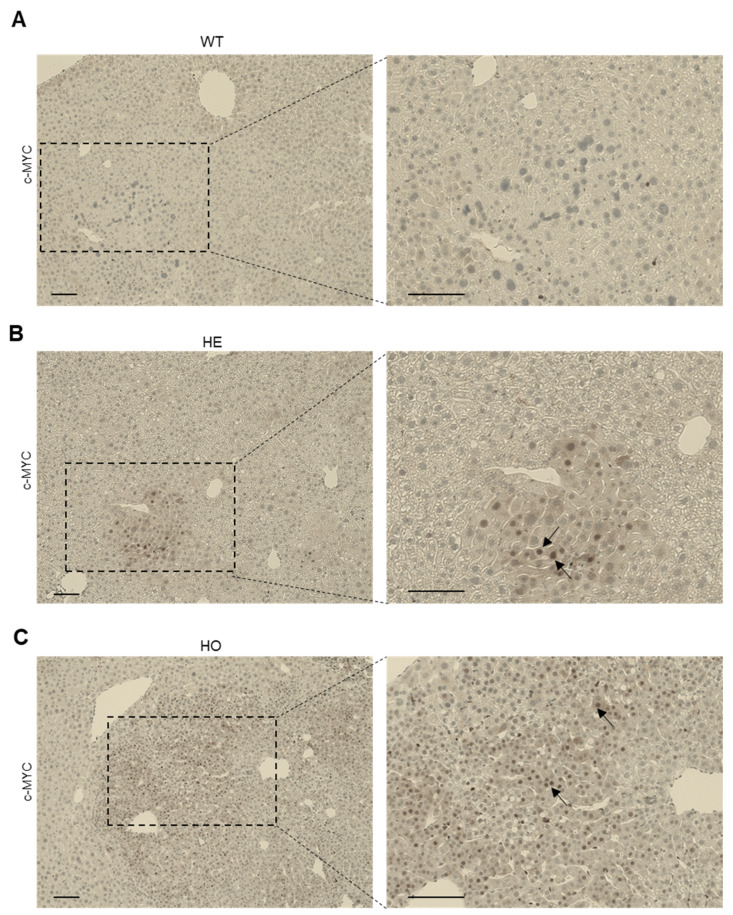
Immunohistochemical analysis of c-MYC expression in livers of WT, HE, and HO *Ppp2r5d* KO mice at 6 months post-DEN treatment. (**A**–**C**) c-MYC expression was determined by IHC. Representative images of liver sections of WT (**A**), HE (**B**), and HO (**C**) *Ppp2r5d* KO mice. Scale bar represents 100 μm.

**Figure 9 cancers-15-04193-f009:**
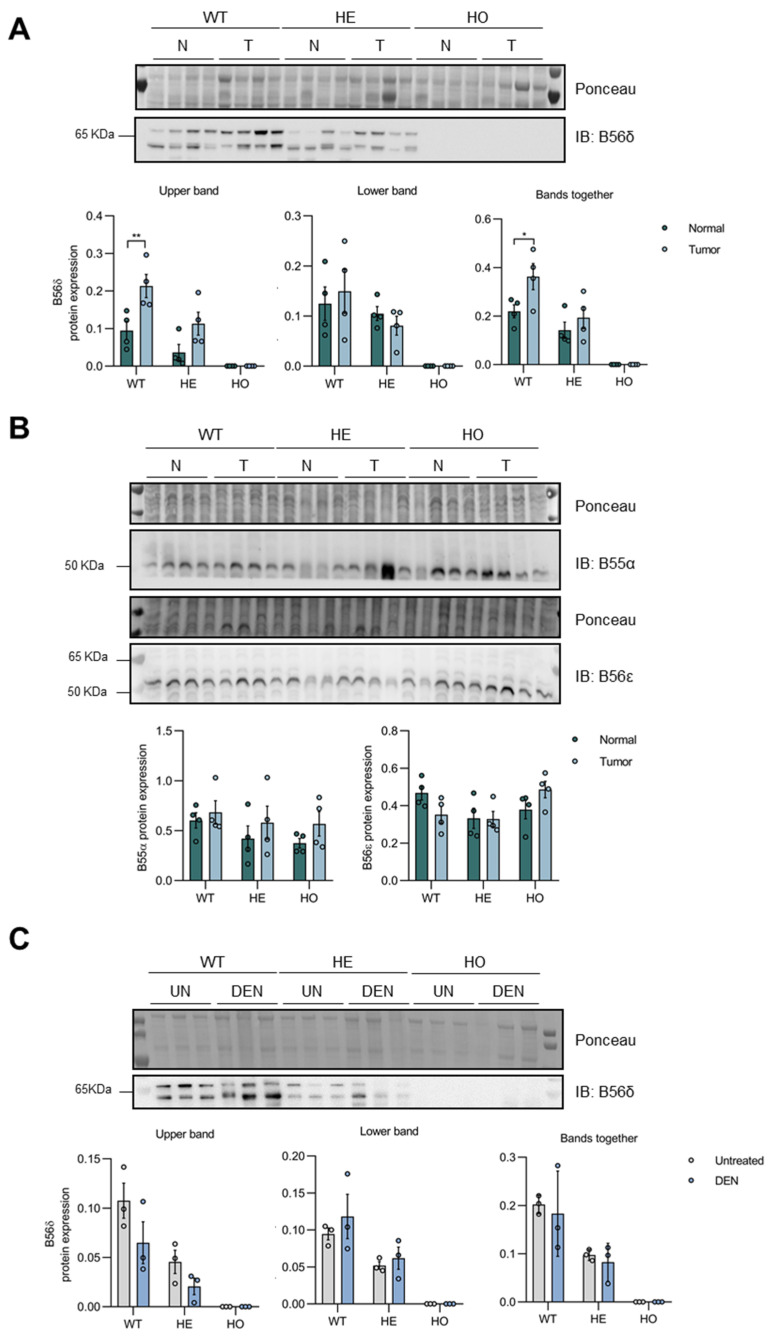
Analysis of B56δ protein expression in WT, HE, and HO *Ppp2r5d* KO mice livers at 11 months post-DEN administration. Protein extracts were prepared from DEN-treated livers of WT, HE, and HO mice. Equal amounts of lysate were separated by SDS-PAGE and subjected to immunoblotting with the indicated antibodies. Ponceau was used for normalization. (**A**) B56δ protein expression was assessed in non-tumor (Normal, N) and tumor liver (T) tissue of 11 months post-DEN WT, HE, and HO *Ppp2r5d* KO mice. (**B**) B55α and B56ε protein expression was assessed in non-tumor (Normal, N) and tumor liver (T) tissue of 11 months post-DEN WT, HE, and HO *Ppp2r5d* KO mice. (**C**) B56δ protein expression was assessed in untreated (UN), and 11 months DEN-treated liver (DEN) tissues of WT, HE, and HO *Ppp2r5d* KO mice. Dots represent individual measurements within a group. Data are represented as mean ± S.E.M. (* *p* < 0.05, ** *p* < 0.01). Two-way ANOVA analysis was used. GraphPad Prism 8.0. See Appendix A for the original image of Western Blots.

## Data Availability

The data presented in this study are available in this article (and Appendix A).

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
