# Peer review of "A Novel Mouse Model of Combined Hepatocellular-Cholangiocarcinoma Induced by Diethylnitrosamine and Loss of Ppp2r5d"

_cancers, 2023, doi:10.3390/cancers15164193_

Round 1

Reviewer 1 Report

Omella et al. have reported a delicately created hepatocellular and cholangiocarcinoma mice model through the knockout of Ppp2r5d gene in combination with the induction using diethylnitrosamine. They have illustrated the role of Ppp2r5d gene in the regulation of oncogenic genes such as c-Myc, while Ppp2r5d is part of a negative feedback induced by c-Myc pathway with a potential role to inhibit oncogenesis. This work provided potential therapeutic pathways in hepatocellular carcinoma and cholangiocarcinoma. However, the reviewer do have a few comments that the authors could address and some suggestions to increase the significance of clinical relevance.

Major

1)    This work could illustrate novel pathways and therapeutics in hepatic tumors that still present unmet needs from clinical perspectives. The reviewer would suggest that the authors bring more clinical relevance to this work. Using a cancer genomic database to perform analyses and support the authors’ conclusion can bring more significance since the authors are proposing a new pathway. A search on cBioPortal liver cancer databases for the expression /amplification of PPP2R5D, as well as using the co-expression tab can well supplement with real-world data instead of just pure in-vivo results. For example https://www.cbioportal.org/results/coexpression?plots_horz_selection=%7B%7D&plots_vert_selection=%7B%7D&plots_coloring_selection=%7B%7D&gene_list=PPP2R5D&cancer_study_list=pancan_pcawg_2020&case_set_id=all&profileFilter=mutations%2Cstructural_variants%2Ccna%2Cgistic&RPPA_SCORE_THRESHOLD=2.0&Z_SCORE_THRESHOLD=2.0&geneset_list=%20

2)    The background of the western blots is over-adjusted. It is much appreciated that the authors showed the ponceau staining of the membrane. However, cutting up the membrane for antibody incubation is more considered a questionable practice these days. Also, based on the original images of the membranes, there are too many unspecific bands. Since the western blot results are the core of this work. The authors should refine the western blot, e.g., the AKT and YAP membranes.

Minor

1)    Line 553: there is no base to claim that the smaller band is a proteolytic fragment of the original protein.

Author Response

Reviewer 1:

Omella et al. have reported a delicately created hepatocellular and cholangiocarcinoma mice model through the knockout of Ppp2r5d gene in combination with the induction using diethylnitrosamine. They have illustrated the role of Ppp2r5d gene in the regulation of oncogenic genes such as c-Myc, while Ppp2r5d is part of a negative feedback induced by c-Myc pathway with a potential role to inhibit oncogenesis. This work provided potential therapeutic pathways in hepatocellular carcinoma and cholangiocarcinoma. However, the reviewer does have a few comments that the authors could address and some suggestions to increase the significance of clinical relevance.

Major

  • This work could illustrate novel pathways and therapeutics in hepatic tumors that still present unmet needs from clinical perspectives. The reviewer would suggest that the authors bring more clinical relevance to this work. Using a cancer genomic database to perform analyses and support the authors’ conclusion can bring more significance since the authors are proposing a new pathway. A search on cBioPortal liver cancer databases for the expression /amplification of PPP2R5D, as well as using the co-expression tab can well supplement with real-world data instead of just pure in-vivo results. For example:

https://www.cbioportal.org/results/coexpression?plots_horz_selection=%7B%7D&plots_vert_selection=%7B%7D&plots_coloring_selection=%7B%7D&gene_list=PPP2R5D&cancer_study_list=pancan_pcawg_2020&case_set_id=all&profileFilter=mutations%2Cstructural_variants%2Ccna%2Cgistic&RPPA_SCORE_THRESHOLD=2.0&Z_SCORE_THRESHOLD=2.0&geneset_list=%20

Author response:

We totally agree with the reviewer about the importance of bringing the clinical relevance of our study to the attention of the readers of our paper. That is why we actually have a follow-up paper ready for submission, solely focused on the role of PPP2R5D in the HUMAN liver. In this follow-up manuscript, we comprehensively describe the status of PPP2R5D together with all other ‘PP2A’ genes in human HCC tumors, and identified a new mechanism of PP2A-B56δ inhibition, which does not involve a genomic alteration of PPP2R5D itself (i.e. genetic alterations and/or downregulation of PPP2R5D). We also confirm the tumor suppressive role of PPP2R5D in different human cell models, in vitro and in vivo. Obviously, presenting all these data falls beyond the scope of the current manuscript, which focuses on describing a new murine model of cHCC-CCA. Nevertheless, to still accommodate the reviewer's comment, we now mention and refer the work of Cunningham et al., who found PPP2R5D amplification/upregulation in several human cancers, including HCC (PMID: 27557495), very much in line with our current data and the follow-up manuscript (line 643-644, in the discussion).

  • The background of the western blots is over-adjusted. It is much appreciated that the authors showed the ponceau staining of the membrane. However, cutting up the membrane for antibody incubation is more considered a questionable practice these days. Also, based on the original images of the membranes, there are too many unspecific bands. Since the western blot results are the core of this work. The authors should refine the western blot, e.g., the AKT and YAP membranes.

Author response:

We apologize for the confusion caused by the bands not related to the antibody (referred to as ‘unspecific bands’). Actually, they are not unspecific bands, but correspond to the signal(s) of previous antibody incubations on the same blot (e.g. ERK antibody). To clarify this, we have now indicated the antibody next to the band in the uncropped blots (see Figures S3 to S9).

It is important to mention that the bands were quantified prior to any possible adjusted background. The adjusted background was only applied in the images found in the manuscript to make the interpretation of the Western blots clearer for the reader. Nevertheless, all samples/conditions were loaded on the same gel, and incubated simultaneously with the antibody. Thus, all samples were exposed to the same background/exposure time, which should not affect the results. When the molecular weight of the proteins was considered significant enough, we cut the membranes to perform several incubations simultaneously, thereby saving material (samples were limited), and time.

Minor

Line 553: there is no base to claim that the smaller band is a proteolytic fragment of the original protein.

Author response:

We are sure that the smaller band corresponds to some form of B56δ protein, as it could no longer be detected in the homozygous Ppp2r5d KO mice with the B56δ antibody. As explained, we searched in the NIH Gene database for an additional transcript of the Mus musculus Ppp2r5d gene, and we found two transcript variants (isoform1 - NM_009358.3 and isoform2 - NM_001357684.1): however, they only differ by one amino acid following translation, which could not explain the difference in molecular weight seen in our Western blots. Although this would exclude that the smaller band is an alternative transcript, it might still result from alternative translation (instead of partial proteolysis), and we have added this as a second, optional explanation (line 556-561).

Reviewer 2 Report

The present paper describes an interesting finding in a novel mouse model: after different periods of treatment with diethylnitrosamine in mice with homo- or heterozygous knock-out for ppp2r5d, the Authors observed the development of cHCC-CCA, in addition to the expected high-grade HCC. Therefore, they concluded to have found a model for the development of combined primary liver cancers (PLC).

The paper is interesting and well written; figures are adequate, and data well presented. Some issue have to be addressed:

1. The main issue of the paper is that cHCC-CCA seems to be considered as a subgroup of HCC (the latter often referred as "classic HCC"), while according to the 2019 WHO mixed PLC are separate entities, comprising cHCC-CCA and the so-called undifferentiated carcinomas (the old "transitional tumors"). According to previous papers, these cancers have different mutation profiles, sometimes resembling HCC and sometimes CCA (please see and cite PMID: 31374137, PMID: 29976634 and similar). In your particular mouse model, the evolution towards cHCC-CCA with KRAS, AKT etc. mutations seems to make these tumors more similar to CCA than HCC, at least in this case.

2. For the same reason, are you sure that the finding of a poorly differentiated morphology together with "cholangio" markers such as SOX9 in the first HCC could not be due to the fact that they already are undifferentiated PLCs? a mention should be made in the discussion.

Also, in the discussion (page 22, raw 683), it is written that "tumors woth mixed HCC and CCA features represent a more advanced stage of HCC,...". According to 2019 WHO this sentence is not correct: it is true that most mixed PLC are more aggressive than most HCC, but they are defined as a separate entity.

3. There are full sentences without proper references, for example: page 3, raw65-66 "Recent studies have identified HCC tumors with ...", or page 6, raw 236-237 "As commonly seen in tumor derived from a single ...".

4. The detailed results listed in the Figure legends Nr. 1, 3 and 4 should be reported in full in the text, especially the P-values and the odd ratios, in order to make results complete. The statistical differences among study groups should be always reported as well, even if not significant.

5. Starting many sentence with an adverb ("moreover", "in addition", "notably") in the same paragraph renders the text more difficult to read, please remove some.

6. Discussion, page 21, raw 641-644. This is an interesting passage, and I think that the data reported "in support of this hypothesis" should actually be shown. Maybe a supplemental file could provide the data to whom is interested without weighing the text down.

7. The Abstract is fully descriptive and totally devoid of results. At the beginning of the abstract cHCC-CCA is defined as "combined hepatocellular carcinoma", instead of "combined hepatocellular-cholangiocellular carcinoma". 

Author Response

Reviewer 2:

The present paper describes an interesting finding in a novel mouse model: after different periods of treatment with diethylnitrosamine in mice with homo- or heterozygous knock-out for ppp2r5d, the authors observed the development of cHCC-CCA, in addition to the expected high-grade HCC. Therefore, they concluded that to have found a model for the development of combined primary liver cancers (PLC).

The paper is interesting and well-written; the figures are adequate, and the data is well-presented. Some issues have to be addressed:

1) The main issue of the paper is that cHCC-CCA seems to be considered as a subgroup of HCC (the latter often referred as "classic HCC"), while according to the 2019 WHO mixed PLC are separate entities, comprising cHCC-CCA and the so-called undifferentiated carcinomas (the old "transitional tumors"). According to previous papers, these cancers have different mutation profiles, sometimes resembling HCC and sometimes CCA (please see and cite PMID: 31374137, PMID: 29976634 and similar). In your particular mouse model, the evolution towards cHCC-CCA with KRAS, AKT etc. mutations seems to make these tumors more similar to CCA than HCC, at least in this case.

Author response:

We thank the reviewer for bringing to our attention the updated classification of primary liver cancer. We have made several changes throughout the text to accommodate this criticism, as follows:

  • Line 65-70: “Recent studies have identified a third subtype of PLC with tumors comprising both, HCC and CCA morphological features, known as combined HCC-CCA (cHCC-CCA) or “biphenotypic” PLC [14,15, 16]. Importantly, there are two main forms of cHCC-CCA: the classical form with tumors containing typical HCC and CCA areas, and the intermediate cell carcinoma with stem-cell features and composed of intermediate cells [17].
  • References 14 and 15 (PMID: 31374137, PMID: 29976634) correspond to the suggested references by the reviewer, and reference 17 (PMID: 29360137) is an additional one to support the classification of mixed/combined PLC.
  • Line 70-71: Removed “classical” from “These heterogeneous tumors are more aggressive and have a poorer prognosis than classical HCC”; and line 691-692: replaced 'classical' by 'typical'
  • Line 695-698: “other studies have suggested that de- or transdifferentiation of HCC-like hepatocytes may be the source of the CCA component in cHCC-CCA”

2) For the same reason, are you sure that the finding of a poorly differentiated morphology together with "cholangio" markers such as SOX9 in the first HCC could not be due to the fact that they already are undifferentiated PLCs? a mention should be made in the discussion.

Also, in the discussion (page 22, raw 683), it is written that "tumors with mixed HCC and CCA features represent a more advanced stage of HCC,...". According to 2019 WHO this sentence is not correct: it is true that most mixed PLC are more aggressive than most HCC, but they are defined as a separate entity.

Author response:

  • We found hepatocytes positive for the stem/progenitor marker SOX9 in tumors from wild-type, heterozygous, and homozygous mice (9 and 11 months post-DEN). However, these tumors were negative for the CCA marker CK19, and biliary/progenitor marker EpCAM, concluding that these tumors correspond to less differentiated HCC due to liver injury and not to intermediate cell carcinomas (undifferentiated PLC). We addressed this question in the discussion section line 714-718: “.We found SOX9 positive hepatocytes in tumors from 9 and 11 months post-DEN mice regardless of the genotype, indicative of less differentiated hepatocytes. In contrast, these hepatocytes were negative for EpCAM and CK19, cholangiocyte or stem/progenitor cell markers. Furthermore, morphological characterization of these hepatocytes suggests these tumors are undifferentiated HCCs [17]”.
  • The following sentences were modified to clarify that cHCC-CCA is not an aggressive subtype of HCC upon progression, but that it is an entity by itself:
  1. Line 691: “In human patients and mouse models [91], tumors with mixed HCC and CCA features (combined PLCs) are more aggressive than typical HCCs,…”
  2. Line 576: “Surprisingly, we also observed the formation of cHCC-CCA tumors in older mice.”
  3. Line 288: “3.2. Analysis of livers at 9 and 11 months post-DEN injection highlights the onset and progression of a combined HCC-CCA phenotype specifically in homozygous and heterozygous Ppp2r5d KO mice”
  4. Line 41: “Remarkably, in older mice, Ppp2r5d deletion resulted in cHCC-CCA development in this model, with the CCA component showing increased expression of progenitor markers (SOX9, EpCAM).”

3) There are full sentences without proper references, for example: page 3, raw65-66 "Recent studies have identified HCC tumors with ...", or page 6, raw 236-237 "As commonly seen in tumor derived from a single ...".

Author response:

  • Line 67: In the sentence "Recent studies have identified a third subtype of PLC with tumors...", references 14, 15, and 16 were added.
  • Line 242: In the sentence "As commonly seen in tumor derived from a single ...". reference 45 was added.

4) The detailed results listed in the Figure legends Nr. 1, 3 and 4 should be reported in full in the text, especially the P-values and the odd ratios, in order to make results complete. The statistical differences among study groups should be always reported as well, even if not significant.

Author response:

The following statistical results were added in the main text:

Figure 1

  • Line 232: Odd ratios and p values of % HCC incidence at 6 months; “(WT vs HE: Odd ratio=7, p=0.236; WT vs HO: Odd ratio=16.2 (p=0.076)”
  • Line 236: “(average number of nodules: WT=0.25, HE=1.86, HO=3.5)”
  • Line 239: “(average % tumor area: WT=0.22, HE=0.78, HO=0.85)”

Figure 3

  • Line 300: Sirius red; “(6m vs 11m WT p<0.0001, HE p<0.0001, HO p<0.0001; 9m vs 11m WT p=0.003, HE p=<0.0001, HO p<0.0001)”
  • Line 307: Ki67 p values at 6 months; “(WT vs HE p=0.011; WT vs HO p=0.0194)”

Figure 4

  • Line 317: Odd ratio and p value of % cHCC-CCA incidence at 9 months “(WT vs HE: Odd ratio=4, p=0.236)”
  • Line 327: Odd ratio and p value of % cHCC-CCA incidence at 9 months “(WT vs HO: Odd ratio=6.7, p=0.167)”
  • Line 387: Odd ratio and p value of % cHCC-CCA incidence at 11 months (WT vs HE: Odd ratio=99, p<0.0001)
  • Line 394: Odd ratio and p value of % cHCC-CCA incidence at 11 months (WT vs HO: Odd ratio=99, p<0.0001)

5) Starting many sentences with an adverb ("moreover", "in addition", "notably") in the same paragraph renders the text more difficult to read, please remove some.

Author response:

  • Line 95: Removed “Interestingly” from “Interestingly, in Ppp2r5d KO mice, devoid of the regulatory PP2A-B56δ subunit in all tissues,…”
  • Line 122: Removed “Moreover” from “Livers from mice at 6, 9 and 11 months post-DEN injection were macroscopically, histologically and (immuno)histochemically characterized, moreover and putative alterations in oncogenic
  • Line 250: Removed “Furthermore” from “Furthermore, Tumor cells also showed some steatosis (Figure 2B)”
  • Line 388: Removed “Moreover” from “Moreover, bipotent progenitor cells were seen in continuity…”
  • Line 390: Removed “In addition” from “In addition, cHCC-CCAs nodules showed immunoreactivity for SOX9”
  • Line 449: Removed “Moreover” from “Moreover, several reports have shown the direct regulation of AKT phosphorylation by PP2A”
  • Line 519: Removed “Moreover” and Additionally from “Moreover, the complete or partial Ppp2r5d depletion further promotes YAP activation, whereas. Additionally, complete Ppp2r5d depletion results in the upregulation of AKT activity already in pre-malignant livers”
  • Line 563: Removed “Notably” from “Notably, increment for total B56δ levels was displayed specifically in the tumor in livers from WT and HE mice as opposed to the non-tumoral surrounding tissues (Figure 9A)”
  • Line 564: Removed “Moreover” from “Moreover, this increase was specific for the PP2A B56δ subunit”
  • Line 667: Removed “In addition” from “In addition, in the pre-malignant livers from 11 months post-DEN HE and HO mice,…”
  • Line 685: Remove “Interestingly” from “ Interestingly,in a mouse model of cholangiocarcinogenesis, (tumorigenesis) was impaired after c-MYC deletion

6) Discussion, page 21, raw 641-644. This is an interesting passage, and I think that the data reported "in support of this hypothesis" should actually be shown. Maybe a supplemental file could provide the data to whom is interested without weighing the text down.

Author response:

We included an additional figure in supplementary data (Figure S11) with the Ppp2r5d promotor sequence of Mus musculus indicating the location of the E-box sequences upstream of the transcription start site.

7) The Abstract is fully descriptive and totally devoid of results. At the beginning of the abstract cHCC-CCA is defined as "combined hepatocellular carcinoma", instead of "combined hepatocellular-cholangiocellular carcinoma". 

Author response:

  • We actually consider the abstract to be clear and to contain all the relevant results for the reader. Nevertheless, we detailed an additional result in line 43: “Finally, we observed an upregulation of Ppp2r5d in tumors from wildtype and heterozygous mice, revealing a…”.
  • We corrected combined hepatocellular-cholangiocarcinoma in line 32, as suggested.
  • At second thought, we suspect that the reviewer’s comment might actually have pertained to the ‘Simple Summary’ instead of the ‘Abstract’ (?) This simple summary is indeed totally descriptive and rather mentioning overall conclusions instead of individual results.

Round 2

Reviewer 1 Report

Comments have been well addressed. 

Reviewer 2 Report

The Authors satisfactorily answered to all the issues.

I recommend publication.